# Differential Impact of Valproic Acid on *SLC5A8*, *SLC12A2*, *SLC12A5*, *CDH1*, and *CDH2* Expression in Adult Glioblastoma Cells

**DOI:** 10.3390/biomedicines12071416

**Published:** 2024-06-25

**Authors:** Milda Juknevičienė, Ingrida Balnytė, Angelija Valančiūtė, Marta Marija Alonso, Aidanas Preikšaitis, Kęstutis Sužiedėlis, Donatas Stakišaitis

**Affiliations:** 1Department of Histology and Embryology, Medical Academy, Lithuanian University of Health Sciences, 44307 Kaunas, Lithuania; milda.jukneviciene@lsmu.lt (M.J.); ingrida.balnyte@lsmu.lt (I.B.); angelija.valanciute@lsmu.lt (A.V.); 2Department of Pediatrics, Clínica Universidad de Navarra, University of Navarra, 31008 Pamplona, Spain; mmalonso@unav.es; 3Centre of Neurosurgery, Clinic of Neurology and Neurosurgery, Faculty of Medicine, Vilnius University, 03101 Vilnius, Lithuania; aidanas.preiksaitis@santa.lt; 4Laboratory of Molecular Oncology, National Cancer Institute, 08660 Vilnius, Lithuania; kestutis.suziedelis@nvi.lt

**Keywords:** valproic acid, adult glioblastoma, U87 MG cell, T98G cell, SLC5A8, NKCC1, KCC2, *CDH1*, *CDH2*

## Abstract

Valproic acid (VPA) has anticancer, anti-inflammatory, and epigenetic effects. The study aimed to determine the expression of carcinogenesis-related *SLC5A8*, *SLC12A2*, *SLC12A5*, *CDH1,* and *CDH2* in adult glioblastoma U87 MG and T98G cells and the effects of 0.5 mM, 0.75 mM, and 1.5 mM doses of VPA. RNA gene expression was determined by RT-PCR. *GAPDH* was used as a control. U87 and T98G control cells do not express *SLC5A8* or *CDH1. SLC12A5* was expressed in U87 control but not in T98G control cells. The *SLC12A2* expression in the U87 control was significantly lower than in the T98G control. T98G control cells showed significantly higher *CDH2* expression than U87 control cells. VPA treatment did not affect *SLC12A2* expression in U87 cells, whereas treatment dose-dependently increased *SLC12A2* expression in T98G cells. Treatment with 1.5 mM VPA induced *SLC5A8* expression in U87 cells, while treatment of T98G cells with VPA did not affect *SLC5A8* expression. Treatment of U87 cells with VPA significantly increased *SLC12A5* expression. VPA increases *CDH1* expression depending on the VPA dose. *CDH2* expression was significantly increased only in the U87 1.5 mM VPA group. Tested VPA doses significantly increased *CDH2* expression in T98G cells. When approaching treatment tactics, assessing the cell’s sensitivity to the agent is essential.

## 1. Introduction

Glioblastoma (GBM) has a poor prognosis due to treatment resistance, high relapse rates, and mortality [1]. Valproic acid (VPA) is being investigated as an adjuvant for GBM in combination with chemotherapy and radiotherapy [2,3]. Studies examining the potential of VPA at the beginning of chemoradiotherapy or after chemoradiotherapy to enhance the antineoplastic activity of chemotherapy in GBM patients have shown conflicting results, with the inclusion of VPA in the regimen improving median overall survival [4,5]. In contrast, other data have shown no effect [6,7]. The combination of temozolomide (TMZ), radiotherapy, and high doses of VPA (25 mg/kg/day) treatment in the adult GBM patient population revealed groups with different proteomic characteristics compared to those treated with TMZ and radiotherapy. At the same time, clinical factors showed no association with the effect of the VPA combination [8]. 

The effects of VPA on GBM cells are consistent with biological mechanisms: It is an inhibitor of HDACs [9] and induces apoptosis [10]. Non-toxic concentrations of VPA sensitized U87 and T98G glioma cells to gefitinib, inhibiting cell growth and inducing autophagy through increased formation of intracellular reactive oxygen species [3]. VPA increases the effectiveness of radiotherapy by sensitizing GBM cells [2] and inducing apoptotic responses to irradiation [11]. Short-term treatment with VPA induced a change in the methylation status of O6-methylguanine-DNA methyl-transferase (MGMT), which can be used to sensitize GBM cells and glioblastoma stem cells to TMZ [12,13]. The heterogeneous behavior of GBM stem cell lines in terms of pro-differentiation capacity and changes in DNA methylation during TMZ treatment reflects the heterogeneity of GBM [12]. The effect of VPA on eradicating the stem cell subpopulation is vital for the effective treatment of GBM. Differentiation-promoting and epigenetic therapies are promising approaches to overcome GBM [13]. The inflammatory microenvironment of the GBM tumor, the released cytokines and chemokines, and the activation of inflammatory signaling pathways promote tumor aggressiveness and resistance to treatment. New data on the GBM inflammatory microenvironment are essential for a prospective approach to GBM treatment [14]. VPA has immunomodulatory and anti-inflammatory effects in exposure [15] that may also be related to sex-related differences in VPA metabolism in animals and humans [16,17]. Elucidating the evolution of GBM sex-linked dimorphism and the efficacy of treatments will be essential to improve the effectiveness of treatment and patient survival, and ensuring that personalized treatment based on specific molecular mechanisms of GBM is an essential challenge for further research [18]. Treatment with a combination of VPA and dichloroacetate significantly increased *Slc5a8* gene expression and showed a significant anti-inflammatory effect on thymocytes from male mice [19], and treatment of T lymphocytes from males and females with this combination showed a significant anti-inflammatory effect and gender-related differences [20]. It was reported that different VPA effect the expression of the *SLC12A2* (NKCC1) and *SLC12A5* (KCC2) co-transporter genes in pediatric glioblastoma PBT24 (boys) and SF8628 (girls) cells [21]. The molecular and clinical role of cation-chloride co-transporters illustrates the significant association of KCC2 and NKCC1 with tumorigenesis. It may be necessary for molecular diagnostics and new treatment strategies for cancer patients [22].

When investigating the efficacy of VPA in combination with other drugs, it is also essential to consider potential drug–drug interactions. The results of studies on the effectiveness of VPA are contradictory. VPA can induce a genomic DNA methylation profile that increases susceptibility to VPA but not TMZ [12,13]. VPA-treated GBM cells secreted high amphiregulin levels, whose expression was positively correlated with resistance to TMZ of different GBM cells [23]. VPA induces the activation of the Na^+^-K^+^-2Cl^–^ co-transporter (NKCC1), significantly increasing the expression of the NKCC1 gene (*SLC12A2*) in PBT24 but not affecting SF8628 cells [21]. The TMZ caused significantly increased RNA expression of the *SLC12A2* in both PBT24 and SF8628 cell types [24]. NKCC1 activity is directly related to GBM cell proliferation [25], and increased NKCC1 protein expression in human GBM is associated with tumor grade [26]. The NKCC1 activation is associated with protein phosphorylation of WNK kinases [27,28]. Thus, it is plausible that combining VPA with TMZ could synergistically activate NKCC1 and reduce treatment efficacy.

K-Cl co-transporter (SLC12A5; KCC2), whose expression is reduced in GBM cells, is associated with intracellular ions’ balance. Increased expression of SLC12A5 inhibits GBM cell proliferation [29]. VPA significantly increases the expression of the SLC12A5 gene (*SLC12A5*) in GBM cells; i.e., it promotes the efflux of K^+^ and Cl^–^ ions from the cell, but this effect depends on different GBM cells [21] and does not have an association with DNA methylation [29]. Thus, SLC12A5 may become an important new GBM biomarker of prognostic significance.

Solute carrier family 5 member A8 (SLC5A8) is a sodium (Na^+^) and chloride (Cl^–^) ion-dependent and Na^+^-coupled monocarboxylate co-transporter [30], the activity of which may therefore be dependent on the intracellular Na^+^ and Cl^–^ levels. SLC5A8 is a tumor growth suppressor in primary human and experimental animal gliomas that contributes to carcinogenesis and is repressed by epigenetic mechanisms [31]. The expression of *SLC5A8* in cancer cells is silenced by hypermethylation, and the gene silencing of SLC5A8 by hypermethylation is associated with poor prognosis [30]. VPA can increase the *SLC5A8* expression in GBM cells [21,32]. SLC5A8 induces cell apoptosis via mitochondrial pyruvate-dependent HDAC inhibition [33]. Studies on the co-activity and interaction of the SLC5A8 co-transporter with NKCC1 and KCC2 and their activity may indicate a link between changes in intracellular Na^+^, K^+^, and Cl^–^ ion concentrations in GBM cells and the treatment effect of the drug on GBM cell ion homeostasis. There are almost no studies on the relationship between SLC5A8 co-transporter activity and the regulation of intracellular Na^+^, K^+^, and Cl^–^ levels.

Cadherin E (*CDH1*) and cadherin N (*CDH2*) are significant contributors to tumor development: a form of metaplasia known as epithelial–mesenchymal transition (EMT) [34]. During EMT, epithelial *CDH1* expression is reduced in exchange for increased mesenchymal *CDH2* expression [35]. *CDH1* expression is rare or absent in gliomas, and expression decreases with brain tumor grade [35,36]. *CDH2* is expressed in brain GBM and plays an important role with NKCC1 in glioma genesis [34,37].

There is significant evidence that the expression of specific genes is altered after VPA treatment. However, the relationship between differentially expressed mRNA and protein of the same gene is inconsistent. On a genome-wide scale, the correlation between mRNA and protein is low [38,39,40]. Therefore, it is justified to limit gene expression studies to determining expression only. This study aimed to investigate the effect of VPA on the expression of the NKCC1, KCC2, and SLC5A8 co-transporters genes and the *CDH1* and *CDH2* genes in adult glioblastoma U87 MG (female) and T98G (male) cells. The studies showed differences in the effect of VPA on the expression of the genes studied in U87 MG and T98G cells, and this effect was dose-dependent.

## 2. Materials and Methods

### 2.1. Glioblastoma Cells and The Tested Groups 

The glioblastoma cell line cells of an adult Caucasian 44-year-old female’s high-grade glioblastoma U87 MG cell line (U87; ECACC 89081402), donated by Dr. Arūnas Kazlauskas (Laboratory of Neuro-Oncology and Genetics, Neuroscience Institute, Lithuanian University of Health Sciences, LT-50009 Kaunas, Lithuania), and an adult Caucasian 61-year-old male’s high-grade glioblastoma T98G cell line cells (product code ATTC-CRL-1690), donated by Prof. M.M. Alonso (University of Navarra, Pamplona, Spain), for the study were used.

The U87 cells were cultivated in Dulbecco’s Modified Eagle Medium (DMEM; Gibco, Paisley, UK) supplemented with 10% fetal bovine serum (FBS; Gibco, Paisley, UK) and 1% 100 IU/mL of penicillin and 100 µg/mL of streptomycin (P/S; Gibco, Grand Island, NY, USA), as reported [41]. The T98G cells were cultivated in Advanced Minimum Essential Medium (AMEM; Gibco, Grand Island, NY, USA) and supplemented with 5% FBS, 4 mM of L-glutamine (Glutamax; Gibco, Paisley, UK) and 1% P/S, as described in the product sheet [42].

Then, 10 μL of the tested cells suspension mixed with 10 μL trypan blue solution (Sigma-Aldrich, Irvine, UK) was used to count the cells number in a Neubauer hemocytometer chamber (Brand GmbH + CO KG, Wertheim, Germany). Next, 0.5 × 10^6^ U87 and 0.7 × 10^6^ T98G cells were seeded in 75 cm^2^ vented culture flasks (ThermoScientific, Rochester, NY, USA) with 15 mL of culture media at 37 °C in a 95% O_2_ and 5% CO_2_ atmosphere. After a 24 h incubation, the culture media were changed to media containing VPA (for the groups treated with 0.5 mM or 0.75 mM or 1.5 mM VPA) and or media without VPA (control groups). U87 and T98G cells were treated for 24 h. There were eight tested groups: (1) U87 control (n = 6), (2) U87 treated with 0.5 mM VPA (n = 6), (3) U87 treated with 0.75 mM VPA (n = 6), and (4) U87 treated with 1.5 mM VPA (n = 5) and (5) T98G control (n = 6), (6) T98G treated with 0.5 mM VPA (n = 6), (7) T98G treated with 0.75 mM VPA (n = 6), and (8) T98G treated with 1.5 mM VPA (n = 5). The effect of VPA on the cells was assessed by comparing the control with the treatment with three different concentrations of VPA (Sigma Aldrich, St. Louis, MO, USA) in the medium. 

### 2.2. RNA Extraction and Quantitative Real-Time PCR Analysis

Total RNA was extracted with TRIzol Plus RNA Purification Kit (Life Technologies, Carlsbad, CA, USA). The RNA purity and concentration were assessed using a NanoDrop2000 spectrophotometer (Thermo Scientific, Branchburg, NJ, USA). The RNA integrity was analyzed using an Agilent 2100 Bioanalyzer (Agilent Technologies, Santa Clara, CA, USA) with an Agilent RNA 6000 Nano Kit (Agilent Technologies, Santa Clara, CA, USA). Reverse transcription was performed with 100 ng RNA using Biometra TAdvanced thermal cycler (Analytik Jena AG, Jena, Germany) with the High-Capacity cDNA Reverse Transcription Kit with RNase Inhibitor (Applied Biosystems, Waltham, MA, USA), according to manufacture instructions. The relative RNA expression assay was performed using Applied Biosystems 7900 Fast Real-Time PCR System (Applied Bio-Systems, Waltham, MA, USA) with TaqMan assays (Applied Biosystems, Pleasanton, CA, USA), according to the manufacturer’s recommendations. The reactions were run in triplicates with 4 μL of cDNA template in a 20 μL reaction volume (10 μL of TaqMan Universal Master Mix II, no UNG (Applied Biosystems, Vilnius, Lithuania), 1 μL of TaqMan Gene Expression Assay 20x (Applied Biosystems, Pleasanton, CA, USA), and 5 μL of nuclease-free water (Invitrogen, Paisley, UK) with the program running at 95 °C for 10 min, followed by 45 cycles of 95 °C for 15 s and 60 °C for 1 min. The investigated genes were *SLC12A5* (Hs00221168_m1; 80 bp), *SLC12A2* (Hs00169032_m1; 97 bp), *SLC5A8* (Hs00377618_m1; 88 bp), *CDH1* (Hs01023894_m1; 61 bp), and *CDH2* (Hs00983056_m1; 66 bp). As a control, we used the *GAPDH* (Hs02786624_g1; 157 bp) gene [43,44]. We used CT cut-off at 35 values as described by the others [45], and these values were not used for calculations.

### 2.3. Statistical Analysis

The statistical analysis was performed using IBM SPSS Statistics 29 software. For the relative gene expression study, the Livak (2**^−^**^ΔΔCT^) method [46] was used to calculate the expression between the VPA-treated (test) and control groups of the target genes. The Shapiro–Wilk test was used to verify the normality assumption. The difference between the two independent groups was evaluated using the nonparametric Mann–Whitney *U* test. Significant differences were considered at the value of *p* < 0.05.

## 3. Results

### 3.1. VPA Treatment Effect on SLC5A8 Expression in U87 and T98G Cells

Table 1 shows the *SLC5A8* and *GAPDH* expression data for the U87- and T98G-cell controls and the VPA-treated tested cell groups.

U87 and T98G control cells do not express *SLC5A8*. Treatment of cells with 1.5 mM VPA induced *SLC5A8* expression in U87 cells, but this expression was low. VPA at 0.75 mM and 0.5 mM did not affect the expression of *SLC5A8* in U87 cells, and no gene expression was observed. Treatment of T98G cells with different doses of VPA did not affect *SLC5A8* expression—the gene was not expressed. VPA treatment did not affect the expression of *GAPDH*.

### 3.2. VPA Treatment Effect on SLC12A2 Expression in U87 and T98G Cells

Table 2 shows the *SLC12A2* and *GAPDH* expression data for the tested U87- and T98G-cell groups.

In all cell groups tested, the expression of *GAPDH* was not different between control and cells treated with different doses of VPA; i.e., VPA treatment did not affect the expression of the control gene. 

The *SLC12A2* expression in U87 control cells was significantly lower than in T98G control cells. VPA treatment did not affect *SLC12A2* expression in U87 cells; no significant differences in *SLC12A2* expression were found when comparing the U87-cell groups tested.

The T98G cells treated with 0.75 mM VPA had significantly higher *SLC12A2* expression than controls, and the relative expression of *SLC12A2* was 1.71-fold higher than that of T98G controls. The *SLC12A2* expression of the T98G 0.5 mM VPA group was not different from that of the T98G control group and was significantly lower than that of the T98G 0.75 mM VPA group and the T98G 1.5 mM VPA group.

The T98G-cell groups treated with 1.5 mM, 0.75 mM, or 0.5 mM VPA had significantly higher *SLC12A2* expression than the respective U87-cell groups.

### 3.3. The VPA Treatment Effect on SLC12A5 Expression in U87 and T98G Cells

Table 3 shows the *SLC12A5* and *GAPDH* expression data for the U87- and T98G-cell groups.

*SLC12A5* was expressed in U87 controls, and *SLC12A5* was not expressed in T98G cells. Treatment of U87 cells with 1.5 mM, 0.75 mM, or 0.5 mM VPA significantly increased *SLC12A5* expression, and the relative expression of *SLC12A5* was 5.97-, 4.40-, and 4.54-fold higher than that of controls, respectively. No significant difference was found when comparing the gene expression of the U87 groups treated with different doses of VPA.

No expression of *SLC12A5* was detected in T98G control cells and T98G 1.5 mM VPA, T98G 0.75 mM VPA, and T98G 0.5 mM VPA cell groups.

### 3.4. The VPA Treatment Effect on CDH1 Expression in U87 and T98G Cells

Table 4 shows the *CDH1* and *GAPDH* expression data for the U87- and T98G-cell groups. 

*CDH1* was inactive in U87 control cells, and VPA treatment of U87 cells did not affect *CDH1* expression.

*CDH1* expression was detectable in T98G 1.5 mM VPA and T98G 0.75 mM VPA cells, whereas T98G control and T98G 0.5 mM VPA groups did not express the *CDH1* gene.

### 3.5. The VPA Treatment Effect on CDH2 Expression in U87 and T98G Cells

Table 5 shows the *CDH2* and *GAPDH* expression data for the U87- and T98G-cell groups.

T98G control cells showed significantly higher *CDH2* expression than U87 control cells. Compared to U87 control cells, *CDH2* expression was significantly higher in the U87 1.5 mM group with a relative expression of 3.41. There was no difference in *CDH2* expression among control U87, U87 0.75 mM VPA, and U87 0.5 mM VPA groups. U87 1.5 mM VPA cells showed significantly higher *CDH2* expression than U87 0.75 mM VPA cells. U87 cells treated with 1.5 mM and 0.5 mM VPA did not differ in expression.

Compared to controls, all doses of VPA significantly increased *CDH2* expression in T98G cells: There was 2.34-fold higher expression in the T98G 1.5 mM VPA group, 2.11-fold higher expression in the T98G 0.75 mM group, and 1.73-fold higher expression in the T98G 0.5 mM VPA group than in control. A comparison of the VPA-treated T98G groups showed no difference in *CDH2* expression.

A comparison of *CDH2* expression between U87 and T98G cells showed that T98G 1.5 mM VPA, T98G 0.75 mM VPA, and T98G 0.5 mM VPA cells had higher gene expression than the corresponding U87 cells.

## 4. Discussion

The application of integrative approaches that combine data from multiple mechanisms enables us to understand disease pathogenesis and develop diagnostic tools to predict brain cancer’s progression or its treatment’s effectiveness [47]. Elucidating the functions of biomolecules and their interrelationships can help interpret the course of disease. In this study, the gene expression of NKCC1, KCC2, *SLC5A8*, *CDH1*, and *CDH2* genes in GBM cells and the possible effect of VPA on their gene expression were determined. The data allowed us to address the potential interrelationship between the expression of studied specific markers. 

A comprehensive study found that KCC2 and NKCC1 co-transporters have opposing cancer-regulatory mechanisms [22]. High-grade GBM cells are associated with increased intracellular chloride [Cl^−^]i level [26], which is associated with increased Na^+^-K^+^-Cl^−^ co-transporter-1 (NKCC1, encoded by *SLC12A2*) and decreased K^+^-Cl^−^ co-transporter (KCC2, encoded *SLC12A5*) activity [48,49]. The [Cl^−^]i content of high-grade GBM cells is significantly higher than the average in grade II glioma and normal cortical cells [26]. 

DNA methylation is an essential epigenetic mechanism that regulates gene expression. Decreased DNA methylation in gene promoters usually leads to the activation of gene transcription, while increased methylation often inhibits gene expression [50]. Phosphorylation has emerged as a key means to rapidly and reversibly modulate the intrinsic transport activity of NKCC1 and KCC2 as a potential therapeutic effect by regulating [Cl^−^]i levels [51,52]. The activity of NKCC1 is precisely regulated by protein phosphorylation and dephosphorylation, where several kinases have been proposed to regulate NKCC1 expression and activity through phosphorylation determined by the balance between kinase and protein phosphatase activities in the neuronal cells [25,50]. VPA has been shown to induce acetylation and demethylation in the test system, and VPA-induced histone acetylation and DNA demethylation have been shown to activate gene expression [32]. 

Our study showed that the *SLC12A2* expression in U87 control cells was significantly lower than in T98G control cells. VPA treatment did not affect *SLC12A2* expression in U87 cells, whereas VPA treatment increased *SLC12A2* expression in T98G cells dose-dependently. The T98G-cell groups treated with 1.5 mM, 0.75 mM, and 0.5 mM VPA had significantly higher *SLC12A2* expression than the respective U87-cell groups. The differential VPA effect on *SLC12A2* expression in pediatric GBM cells was reported also [21]. 

NKCC1 was shown to be highly expressed in gliomas, and a higher glioma grade directly correlated with NKCC1 expression [37]. NKCC1 is one of the most important transporters of Cl^−^ into cells and regulates cell volume expansion [53]. Increased NKCC1 protein expression in human GBM attenuates cancer cell proliferation and migration, and inhibition of NKCC1 activity impairs tumor invasion and cell apoptosis [25,26,27,54,55,56]. Bioinformatic analysis showed that high glioma NKCC1 expression is associated with MAPK signaling pathways, TGF-beta signaling, and epithelial–mesenchymal transition regulation, and its expression in mesenchymal GBMs was associated with GBM patients’ shorter survival and worse prognosis [37].

Our study data show that the *SLC12A5* was expressed in U87 controls and not in T98G cells. Treatment of U87 cells with VPA significantly increased *SLC12A5* expression, but there was no association with the doses of VPA tested. No expression of *SLC12A5* was detected in the T98G control and T98G VPA-treated cells. Recently, it was reported that VPA differentially but significantly increased *SLC12A5* expression in pediatric GBM cells [21].

The different effects of VPA on KCC2 gene expression in GBM cells that we have identified may be significant in several respects. Cell volume reduction is an early sign of apoptosis associated with the loss of intracellular K^+^ and Cl^−^ anions [29,57], which is associated with caspase activation and caspase cascade-related apoptotic mechanisms [58]. KCC2 positively correlated with the levels of tumor-infiltrating macrophages and CD4^+^ T cells [22]. Bioinformatics analysis suggests that overexpression of *SLC12A5* inhibits the proliferation of glioma U251MG cells, and *SLC12A5* may be a novel effective biomarker of GBM with prognostic significance [29].

Furthermore, in neurons, the regulation of [Cl^−^]i is mediated by NKCC1 and KCC2 transporters: NKCC1 increases, while KCC2 decreases [Cl^−^]i. Histologically reduced neuronal KCC2 staining was reported in adult patients with GBM and epilepsy [49,59]. Alterations in the balance of NKCC1 and KCC2 activity may decrease the hyperpolarizing effects of γ-aminobutyric acid (GABA), contributing to epileptogenesis in human GBM. Associated seizures worsen the prognosis of GBM patients [60,61]. Proper KCC2 activity ensures the functioning of neuronal postsynaptic GABAA receptors by acting as a neuron-specific K^+^ and Cl^−^ extruder, using the K^+^ gradient to maintain low [Cl^−^]i levels. The excitatory effects of GABAA are dependent on relatively high [Cl^−^]i levels [62]. The mechanisms of GBM-associated epilepsy are linked to the reduction of KCC2 activity in the peritumoral region, leading to impaired GABAergic inhibition, and they suggest that modulating [Cl^−^]i homeostasis by activating KCC2 may help control seizures [63]. The drug’s effect on activating the KCC2 function in GBM cells makes it relevant as a potential new anticancer therapeutic target.

The study showed that U87 and T98G control cells do not express *SLC5A8*. Treatment with 1.5 mM VPA induced *SLC5A8* expression in U87 cells, while treatment of T98G cells with VPA did not affect *SLC5A8* expression. SLC5A8 co-transporter is a sodium (Na^+^) and chloride (Cl^−^) ion-dependent and Na^+^-coupled monocarboxylate co-transporter [31,64]. Thus, the activity may depend on the intracellular Na^+^ and Cl^−^ concentration. *SLC5A8* is a tumor growth suppressor in experimental animals and primary human gliomas that contributes to carcinogenesis and is repressed by epigenetic mechanisms [31]. 

The *SLC5A8* is expressed in normal brain cells but is significantly reduced in most human glioma primary cells and cell lines, especially when the associated CpG islands were aberrantly methylated [31]. Hypermethylation silences the expression of *SLC5A8* in cancer cells, and gene silencing of SLC5A8 by hypermethylation was associated with poor prognosis [30]. The reduced expression of *SLC5A8* in the absence of aberrant methylation in a few primary tumors suggests that SLC5A8 may not be affected by multiple epigenetic mechanisms. Ectopic expression of SLC5A8 strongly inhibits colony formation in glioma cell lines, indicating that it suppresses glioma growth in vitro [31]. SLC5A8 induces cell apoptosis via mitochondrial pyruvate-dependent HDAC inhibition [33]. VPA can increase the expression of *SLC5A8* in GBM cells [21,32]. 

SLC5A8 is a transporter that moves short-chain fatty acids and other monocarboxylic acids or drugs, such as pyruvate, butyrate, or dichloroacetate, in a Na^+^-dependent manner [64,65]. Thus, this leads to the hypothesis that SLC5A8-mediated tumor growth inhibition is associated with transposing antiproliferative molecules into the cells, thereby improving mitochondrial function. 

A limitation of our study is that we could not show the interdependence of the studied co-transporters activity. The activity of all co-transporters is attributable to [Cl^−^]i, which can be seen as a signaling pathway. Therefore, it would be meaningful to investigate further the effect of VPA on the mechanisms of [Cl^−^]i regulation in glioblastoma and the malignancy of GBM cells.

The *SLC5A8* expression may also be linked to the activity of the NKCC1 and KCC2 co-transporters function, which, by regulating the Na^+^ and Cl^−^ intracellular levels, may also regulate the Na^+^ and Cl^−^ dependent SLC5A8 co-transporter function. The transport function of SLC5A8 is particularly significant in brain tumors, as butyrate and dichloroacetate are currently being investigated for treating human gliomas [66,67]. 

The data show that *CDH1* was inactive in U87 control and U87 VPA-treated cells. The *CDH1* gene was not expressed in the T98G control and T98G 0.5 mM VPA groups, whereas *CDH1* expression was detectable in T98G 1.5 mM VPA and T98G-0.75 mM VPA cells. The researchers report that *CDH1* expression decreases with brain tumor grade [35]. *CDH1* expression is rare or absent in gliomas; *CDH1* hypermethylation was found in 39.4% of GBM samples [36]. During EMT, epithelial *CDH1* expression decreases and mesenchymal *CDH2* expression increases [35]. Our data on *CDH2* expression showed that *CDH2* expression was significantly higher in T98G control cells than in control U87 cells. *CDH2* expression was significantly increased only in the U87 1.5 mM VPA group. All doses of VPA studied significantly increased *CDH2* expression in T98G cells, whereas *CDH2* expression did not differ. Comparison of *CDH2* expression between U87 and T98G cells showed that T98G cells treated with VPA had significantly higher gene expression than the corresponding U87 cells. In mesenchymal GBMs, *CDH2* is associated with NKCC1 activity and a form of metaplasia, which is termed EMT [34,37]. 

The study showed that the GBM cells tested have different expressions of the genes tested and different effects of VPA on them, which may be an essential avenue for more extensive future studies.

## 5. Conclusions

The expression of *SLC12A2*, *SLC12A5*, and *CDH2* in adult glioblastoma U87 MG and T98G control cells differs significantly. These differences could potentially serve as indicators for assessing tumor malignancy. The studies also revealed distinct responses of the tested cells to VPA treatment, suggesting that the differences in cell marker expression could influence treatment outcomes. This underscores the need for further preclinical and clinical studies on the effect of VPA on tumor cell marker expression, which could open up new approaches for more personalized and effective treatment.

## Figures and Tables

**Table 1 biomedicines-12-01416-t001:** The *SLC5A8* and *GAPDH* expression data from U87- and T98G-cell controls and VPA-treated groups.

Study Group	n	Indicator, Mean ± SD		
		CT		∆CT
		*SLC5A8*	*GAPDH*	
U87 control	6	37.31 ± 1.51	16.25 ± 1.26	–
U87 1.5 mM VPA	5	34.88 ± 0.54	16.73 ± 0.29	18.15 ± 0.57
U87 0.75 mM VPA	6	35.39 ± 1.19	16.41 ± 1.21	–
U87 0.5 mM VPA	6	36.51 ± 0.61	17.87 ± 1.77	–
T98G control	6	not detected	17.41 ± 0.22	–
T98G 1.5 mM VPA	5	36.31 ± 0.82	18.07 ± 0.77	–
T98G 0.75 mM VPA	6	37.16 ± 0.36	18.60 ± 0.37	–
T98G 0.5 mM VPA	6	not detected	17.93 ± 0.76	–

“–“ no gene expression.

**Table 2 biomedicines-12-01416-t002:** The *SLC12A2* and *GAPDH* expression data in the U87- and T98G-cell study groups.

Study Group	n	Indicator, Mean ± SD				
		CT		∆CT	∆∆CT	2^−∆∆CT^
		*SLC12A2*	*GAPDH*			
U87 control	6	22.07 ± 1.23	18.14 ± 1.41	3.93 ± 0.64	0.00 ± 0.64	1.08 ± 0.46
U87 1.5 mM VPA	5	22.62 ± 0.36	18.76 ± 0.93	3.87 ± 0.83	−0.06 ± 0.83	1.20 ± 0.54
U87 0.75 mM VPA	6	22.40 ± 0.36	18.52 ± 1.25	3.88 ± 1.15	−0.05 ± 1.14	1.28 ± 0.71
U87 0.5 mM VPA	6	23.49 ± 1.09	19.72 ± 1.87	3.77 ± 1.16	−0.16 ± 1.16	1.36 ± 0.70
T98G control	6	21.63 ± 0.60	19.87 ± 0.23	1.76 ± 0.60 ^a^	0.00 ± 0.60	1.07 ± 0.19
T98G 1.5 mM VPA	5	21.37 ± 0.55	20.50 ± 0.31	0.87 ± 0.50 ^b^	−0.89 ± 0.50	1.94 ± 0.61
T98G 0.75 mM VPA	6	21.83 ± 0.16	20.81 ± 0.20	1.02 ± 0.30 ^c,d^	−0.75 ± 0.30	1.71 ± 0.35
T98G 0.5 mM VPA	6	22.00 ± 0.26	19.83 ± 0.74	2.17 ± 0.53 ^e,f,g^	0.41 ± 0.53	0.79 ± 0.23

^a^ *p* = 0.002, compared with U87 control; ^b^ *p* = 0.008, compared with U87 1.5 mM VPA; ^c^
*p* = 0.02, compared with T98G control; ^d^ *p* = 0.002, compared with U87 0.75 mM VPA; ^e^ *p* = 0.002, compared with T98G 0.75 mM VPA; ^f^ *p* = 0.02, compared with U87 0.5 mM VPA; ^g^ *p* = 0.004, compared with T98G 1.5 mM VPA.

**Table 3 biomedicines-12-01416-t003:** The *SLC12A5* and *GAPDH* expression data of the studied U87- and T98G-cell groups.

Study Group	n	Indicator, mean ± SD				
		CT		∆CT	∆∆CT	2^−∆∆CT^
		*SLC12A5*	*GAPDH*			
U87 control	6	33.79 ± 1.21	16.25 ± 1.26	17.54 ± 1.07	0.00 ± 1.07	1.05 ± 1.26
U87 1.5 mM VPA	5	31.89 ± 0.66	16.73 ± 0.29	15.16 ± 0.75 ^a^	−2.38 ± 0.75	5.97 ± 3.18
U87 0.75 mM VPA	6	32.00 ± 0.63	16.41 ± 1.21	15.59 ± 0.73 ^b^	−1.95 ± 0.73	4.40 ± 2.39
U87 0.5 mM VPA	6	33.28 ± 1.35	17.87 ± 1.77	15.41 ± 0.45 ^c^	−2.12 ± 0.45	4.54 ± 1.18
T98G control	6	37.31 ± 0.46	17.41 ± 0.22	–	–	–
T98G 1.5 mM VPA	5	36.46 ± 0.60	18.07 ± 0.77	–	–	–
T98G 0.75 mM VPA	6	37.10 ± 0.46	18.60 ± 0.37	–	–	–
T98G 0.5 mM VPA	6	36.79 ± 0.70	17.93 ± 0.76	–	–	–

“–“ no gene expression; ^a^ *p* = 0.002, compared with U87 control; ^b^ *p* = 0.002, compared with U87 control; ^c^ *p* = 0.009, compared with U87 control.

**Table 4 biomedicines-12-01416-t004:** The *CDH1* and *GAPDH* expression data of the studied U87- and T98G-cell groups.

Study Group	n	Indicator, Mean ± SD		
		CT		∆CT
		*CDH1*	*GAPDH*	
U87 control	6	36.21 ± 0.49	16.25 ± 1.26	–
U87 1.5 mM VPA	5	35.86 ± 0.60	16.73 ± 0.29	–
U87 0.75 mM VPA	6	36.41 ± 0.61	16.41 ± 1.21	–
U87 0.5 mM VPA	6	36.12 ± 0.56	17.87 ± 1.77	–
T98G control	6	not detected	17.41 ± 0.22	–
T98G 1.5 mM VPA	5	34.89 ± 0.38	18.07 ± 0.77	16.81 ± 0.57
T98G 0.75 mM VPA	6	34.87 ± 0.38	18.60 ± 0.37	16.27 ± 0.47
T98G 0.5 mM VPA	6	35.08 ± 0.83	17.93 ± 0.76	–

“–“ no gene expression.

**Table 5 biomedicines-12-01416-t005:** The *CDH2* and *GAPDH* expression data of studied U87- and T98G-cell groups.

Study Group	n	Indicator, Mean ± SD				
		CT		∆CT	∆∆CT	2^−∆∆CT^
		*CDH2*	*GAPDH*			
U87 control	6	23.82 ± 0.72	18.14 ± 1.41	5.68 ± 1.26	0.00 ± 1.26	1.01 ± 0.82
U87 1.5 mM VPA	5	22.74 ± 1.44	18.76 ± 0.93	3.98 ± 0.52 ^a^	−1.70 ± 0.52	3.41 ± 1.16
U87 0.75 mM VPA	6	23.60 ± 0.68	18.52 ± 1.25	5.07 ± 0.64 ^b^	−0.61 ± 0.64	1.64 ± 0.62
U87 0.5 mM VPA	6	24.10 ± 1.46	19.72 ± 1.87	4.38 ± 1.11	−1.30 ± 1.11	3.06 ± 1.93
T98G control	6	23.64 ± 0.22	19.87 ± 0.23	3.78 ± 0.30 ^c^	0.00 ± 0.30	1.02 ± 0.17
T98G 1.5 mM VPA	5	23.11 ± 0.46	20.50 ± 0.31	2.61 ± 0.45 ^d,e^	−1.17 ± 0.45	2.34 ± 0.74
T98G 0.75 mM VPA	6	23.52 ± 0.31	20.81 ± 0.20	2.71 ± 0.22 ^f,g^	−1.06 ± 0.22	2.11 ± 0.33
T98G 0.5 mM VPA	6	22.90 ± 0.31	19.83 ± 0.74	3.07 ± 0.55 ^h,i^	−0.71 ± 0.55	1.73 ± 0.55

^a^ *p* = 0.02, compared with U87 control; ^b^ *p* = 0.02, compared with U87 1.5 mM VPA; ^c^ *p* = 0.002, compared with U87 control; ^d^ *p* = 0.008, compared with U87 1.5 mM VPA; ^e^ *p* = 0.004, compared with T98G control; ^f^ *p* = 0.002, compared with T98Gcontrol; ^g^ *p* = 0.002, compared with U87 0.75 mM VPA; ^h^ *p* = 0.04, compared with T98G control; ^i^ *p* = 0.04, compared with U87 0.5 mM VPA.

## Data Availability

The data presented in this study are available on request from the corresponding author.

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
