# Peer review of "Differential Impact of Valproic Acid on SLC5A8, SLC12A2, SLC12A5, CDH1, and CDH2 Expression in Adult Glioblastoma Cells"

_biomedicines, 2024, doi:10.3390/biomedicines12071416_

Round 1
Reviewer 1 Report
Comments and Suggestions for Authors
This manuscript treated adult glioblastoma cells, U87MG ad T98G cells with VPA at different doses to explore the effect of VPA on Slc12A2, Slc12A5, and Slc5A8. The method they used is real time PCR. The whole idea is clear. But the current results are not enough to support the title. I have some question as following: 1. In the introduction part, the other researchers already treated pediatric glioblastoma PBT24 (boys) and SF8628 (girls) cells with VPA and checked the same target, it seems your study is lack of novelty. 2. When CT value is higher than 35, I don't think the result is reliable. I think you need to design new primers to confirm the current results. 3. As you was exploring the impact on the expression, I think western blotting is also needed to check the protein expression. 4. What kind of solvent do you use in the control group? 5. In the method part, you mentioned that "n=6", what this mean?
Comments on the Quality of English Language
The English needs to be improved.
Author Response
Replies to the Reviewer's comments
Thank you for the valuable Reviewer comments. We have corrected and improved the manuscript based on them. Here are the responses to the comments
This manuscript treated adult glioblastoma cells, U87MG ad T98G cells with VPA at different doses to explore the effect of VPA on Slc12A2, Slc12A5, and Slc5A8. The method they used is real time PCR. The whole idea is clear.
But the current results are not enough to support the title.
Answer
Thank you for your comment. The title of the manuscript has been corrected and expanded.
I have some questions as follows:
- In the introduction part, the other researchers already treated pediatric glioblastoma PBT24 (boys) and SF8628 (girls) cells with VPA and checked the same target; it seems your study lacks novelty.
Answer
We agree with the comment. The scope of the studies has been broadened following the comment. We believe that it is appropriate to conduct studies analogous to adult glioblastoma, as the current position is that pediatric glioblastomas are different from adult glioblastomas.
- When CT value is higher than 35, I don't think the result is reliable. I think you need to design new primers to confirm the current results.
Answer
Thank you for a very important comment. We have made corrections in line with the comment. We used a CT cut-off at 35 values. In the previous version, all the thresholds for targets were automated. In the current manuscript version, we have manually changed the threshold for low-expressed genes, as reported in the protocol (TaqMan® Gene Expression Assays Protocol (PN 4333458N) (thermofisher.com)). With these changes, CT has changed, and some have become lower than 35. We have added a "Methods" section accordingly.
- As you was exploring the impact on the expression, I think western blotting is also needed to check the protein expression.
Answer
Your note is important. We believe that the evidence of specific differences in gene expression following VPA treatment is significant. The correlation between differentially expressed mRNA and protein for the same gene is controversial. At the genome-wide scale, the correlation between mRNA and protein is low (1, 2, 3). Thus, the limitations of the protein expression studies for the genes studied may also be based on the likelihood that short-term VPA treatment would be associated with changes in protein expression. We have pointed out that a limitation of the study is that other mechanisms underlying protein activity were not excluded. The data in the study include NKCC1, which determines Cl- influx, and KCC2, which addresses ion leakage within the cell. It would, therefore, make sense to further investigate the effect of VPA on [Cl-]i, which is a signaling pathway that may affect the expression of other genes. However, this could be a follow-up study to assess the effect of VPA on Cl- transport mechanisms.
- de Sousa Abreu R., Penalva L.O., Marcotte E.M., Vogel C. Global Signatures of Protein and MRNA Expression Levels. Mol. Biosyst. 2009;5:1512–1526. doi: 10.1039/b908315d.
- Vogel C., Marcotte E.M. Insights into the Regulation of Protein Abundance from Proteomic and Transcriptomic Analyses. Nat. Rev. Genet. 2012;13:227–232.
- Koussounadis A., Langdon S.P., Um I.H., Harrison D.J., Smith V.A. Relationship between Differentially Expressed MRNA and MRNA-Protein Correlations in a Xenograft Model System. Sci. Rep. 2015;5:10775.
- What kind of solvent do you use in the control group?
Answer
In control groups, we used just growth media as described in the “Material and Methods” section.
- In the method part, you mentioned that "n=6", what this mean?
Answer
n = 6 means that in the study group, we used 6 samples.
We thank the Reviewer for his valuable notes. We hope to have taken them into account and we have revised the manuscript accordingly, which was certainly improved.
Sincerely,
Donatas Stakišaitis

Reviewer 2 Report
Comments and Suggestions for Authors
1) First, the title, the results and the experiments conducted do not justify "personalized therapy".
Abstract:
2) Please provide information in the background section why it is important to know the expression of SLC5A8, SLC12A2 and SLC12A5 in glioblastoma?
3) How the differential response to VPA reflects the biological nature and potential difference in malignancy. Provide further details.
4) The assessment of the sensitivity of the cell to the treatment with the agent is essential when approaching the treatment tactics - it is not the conclusion of the experiments performed. Please be more specific about the results of the work.
Introduction
5) What is the rationale for the study of the effect of VPA on the expression of co-transporter genes?
Materials and Methods
6) Provide more details about the PCR protocol. In addition to the controls used, the number of replicates and experiments performed.
Results
7) Other mechanistic experiments can support the conclusions highlighted in the Abstract.
From using gene silencing and bioinformatic tools to simple cell viability or proliferation assays.
Discussion and Conclusions
8) What is the mechanism of action of VPA treatment?
9) Which pathways are involved in the differential expression of co-transporter genes in these cells?
10) How can treatment be individualized in this context?
Author Response
Replies to the Reviewer's comments
Thank you for the valuable Reviewer’s comments. We have corrected and improved the manuscript based on them. Here are the responses to the comments
1) First, the title, the results and the experiments conducted do not justify "personalized therapy".
Answer
We have corrected the title in line with the comments.
Abstract:
2) Please provide information in the background section why it is important to know the expression of SLC5A8, SLC12A2 and SLC12A5 in glioblastoma?
Answer
The abstract allows a very limited number of words. We have indicated under the note that we investigated the expression of carcinogenesis-related SLC5A8, SLC12A2, SLC12A5, CDH1 and CDH2 in adult glioblastoma U87 MG and T98G cells and the effects of 0.5 mM, 0.75 mM and 1.5 mM doses of VPA.
3) How the differential response to VPA reflects the biological nature and potential difference in malignancy. Provide further details.
Answer
We hope that the data from our study provides some insight into the malignancy of the cells studied or the effect of VPA on it. For example, increased expression of NKCC1 gene may be associated with an unfavorable course or ineffectiveness of treatment. Conversely, increased expression of KCC2, SLC5A8 genes may lead to onco-suppression. The expression of CDH1 and CDH2, or the relationship between their changes, may also be indicative of treatment effects. We believe that we have highlighted these associations to some extent by adding additional information to the article based on the comments.
4) The assessment of the sensitivity of the cell to the treatment with the agent is essential when approaching the treatment tactics - it is not the conclusion of the experiments performed. Please be more specific about the results of the work.
Answer
We have adjusted the conclusions in line with the comment
Introduction
5) What is the rationale for the study of the effect of VPA on the expression of co-transporter genes?
Answer
The NKCC1, KCC2, and SLC5A8 carriers under investigation are important for the apoptosis machinery. Therefore, their gene expression studies and the effect of VPA on expression are important for assessing anticancer effects. The cellular differences identified and the differences in the effect of VPA on the cells studied may be important in the future when deciding on individualized treatment approaches.
Materials and Methods
6) Provide more details about the PCR protocol. In addition to the controls used, the number of replicates and experiments performed.
Answer
Thank you for the notice. We provided more details in Material and Methods section.
Results
7) Other mechanistic experiments can support the conclusions highlighted in the Abstract.
From using gene silencing and bioinformatic tools to simple cell viability or proliferation assays.
Answer
We agree with your important point. However, in this manuscript, we have limited ourselves to gene expression studies. We have also added data from the 1.5 mM VPA exposure studies and CDH1 and CDH2 expression studies. We believe that we have added to the conclusions accordingly. Thank you for your suggestions.
Discussion and Conclusions
8) What is the mechanism of action of VPA treatment?
Answer
We believe that the effect on the expression of the genes studied partly reflects the anti-cancer mechanisms of VPA. Importantly, some of the changes in gene expression may be unfavorable and may be related to individual cellular characteristics, such as an increase in NKCC1 expression. Therefore, studies to establish the rationale for personalized treatment are important. It is also important to emphasize that VPA metabolism is also gender-related, and therefore, research on this aspect is also relevant.
9) Which pathways are involved in the differential expression of co-transporter genes in these cells?
Answer
Thank you for this comment. This comment led us to investigate the link between CDH2 and NKCC1. In mesenchymal GBMs, CDH2 is associated with NKCC1 activity and a form of metaplasia termed EMT. We think that this addition of data to the discussion, showing possible pathways, has really improved the article
10) How can treatment be individualized in this context?
Answer
Our studies are limited to glioma cell lines. But the studies suggest that differences in the efficacy of VPA, and differences in gene expression in the cells, may be important in deciding aspects of personalised treatment. We are continuing our study with primary glioblastoma cells on this basis and we think that this will answer the questions related to this aspect.
We appreciate your comments and advice. They have been important in correcting the shortcomings of the paper and in adding to it.
Sincerely,
Donatas Stakišaitis
Reviewer 3 Report
Comments and Suggestions for Authors
Line 40
TMZ write that this TMZ stand for temozolomida used to glioblastoma and astrocitoma.
Line 106
I wonder how the changes in Na+ K+ and Cl- can modify the membrane potential that can allow more or less cell activity. Taking account that he quantity of ion transporters can modify such membrane property.
Is not clear to me the value of DCT in U87 – control in tables 1, 2, and 3, from where this value is gotten?
However, the reference 35 explain to detail the derivation of the analysis of Real time PCR.
Line 347-348
“Therefore, it would be meaningful to investi-347 gate further the effect of VPA on the mechanisms of [Cl−]i regulation in glioblastoma and 348 the malignancy of GBM cells.”
Should be important to know the changes in membrane potential and the reversal potentials of Cl- and Na+ in order to know the effects of GABA level if inhibition or may be excitation?
I wonder over the quantity of VPA, how high is this concentration?
I consider that other doses can be used to quantify properly the effects.
Author Response
Replies to the Reviewer's comments
Thank you for the valuable Reviewer comments. We have corrected and improved the manuscript based on them. Here are the responses to the comments
Line 40
TMZ writes that this TMZ stands for temozolomide, used for glioblastoma and astrocytoma.
Answer
The abbreviation TMZ stands for temozolomide. We have made a correction according to the comment.
Line 106
I wonder how the changes in Na+ K+ and Cl- can modify the membrane potential that can allow more or less cell activity. Taking account that he quantity of ion transporters can modify such membrane property.
Answer
The stoichiometry of ions carried across the membrane by the NKCC1 cotransporter is neutral with respect to charge. NKCC1 promotes a Cl- accumulation above the value determined by the resting membrane potential and, thus, a depolarized Cl- reversal potential. Increased [Cl-]i could Favor increased the cell volume. The activity of the NKCC1 gene would increase the [Cl-]i.
Is not clear to me the value of DCT in U87 – control in tables 1, 2, and 3, from where this value is gotten? However, the reference 35 explain to detail the derivation of the analysis of Real time PCR.
Answer
Reference 35 explains the Livak method and how the relative expression value is calculated. The ΔCT value is calculated from the target gene CT value minus the reference gene CT value. In Tables 1 and 3 of the current version of the manuscript, we have made corrections to the previous version's calculations of delta CT values in the U87 control cell. We used a CT cutoff value of 35. In the previous version, all target thresholds were automated. In the current version of the manuscript, we manually changed the threshold for low expressed genes as specified in the protocol (TaqMan® Gene Expression Assays Protocol (PN 4333458N) (thermofisher.com)). After these changes, the CTs changed and some became less than 35. We have added the 'Methods' section accordingly. Thank you for your comment and we hope to respond accordingly.
Line 347-348
“Therefore, it would be meaningful to investigate further the effect of VPA on the mechanisms of [Cl−]i regulation in glioblastoma and the malignancy of GBM cells.”
Should it be important to know the changes in membrane potential and the reversal potentials of Cl- and Na+ in order to know the effects of GABA level if inhibition or maybe excitation?
Answer
Thank you for your comment. The activities of NKCC1 and KCC2 are closely correlated. Our data show a heterogeneous effect of VPA on cotransporters gene expression in different cells. So this is an interesting area for further research. The unequal carrier activity or the effect of VPA on [Cl-]i is also related to the activity of Cl- channels, e.g. the GABAA receptor is itself a Cl- channel. Thus, further research into the mechanisms of Cl- transport is relevant and may reveal individual differences in VPA treatment.
I wonder over the quantity of VPA, how high is this concentration?
I consider that other doses can be used to quantify properly the effects.
Answer
We thank you for this comment. It encouraged us to carry out the tests with a 1.5 mM VPA concentration. These studies proved to be important and nicely complemented the second version fixed of the manuscript. Thank you for the idea.
We thank the Reviewer for his valuable remarks and advice. We hope that we have taken this into account and have made appropriate corrections to the manuscript, which has certainly improved.
Sincerely,
Donatas Stakišaitis
Round 2
Reviewer 1 Report
Comments and Suggestions for Authors
I can see the revisions based on the comments. But I can still find many issues need to be corrected.
1. In Table2, I can't find U87-1.5 mM PVA group.
2. I don't think it's necessary to show both Figure 1A and 1B, because they are showing the same thing. And for Figure 1B, the mean for the control group is 1. The Figure Legend also need to be corrected. There are same issues in Figure 2 and 3.
Author Response
Replies to the Reviewer's comments
Thank you for the valuable Reviewer comments. We have corrected and improved the manuscript based on them.
|
Does the introduction provide sufficient background and include all relevant references? Can be improved |
|
Answer |
|
The Introduction section adds a justification for why only gene expression was studied, supplemented by three related references. The numbering of the references has, therefore, changed since the addition of these references (this can be seen by making changes with Track changes). Are the conclusions supported by the results? Can be improved Answer The conclusions have been rewritten. |
Comments and Suggestions for Authors
I can see the revisions based on the comments. But I can still find many issues need to be corrected.
- In Table2, I can't find U87-1.5 mM VPA group.
Answer
Table supplemented with data from the U87-1.5 mM VPA group. Thanks for the important note.
- I don't think it's necessary to show both Figure 1A and 1B, because they are showing the same thing. And for Figure 1B, the mean for the control group is
The Figure Legend also need to be corrected. There are same issues in Figure 2 and 3.
Answer
In Figures 1, 2, and 3, we have removed part A and left only the relative gene expression graph (part B). We assume that the relative gene expression data are not repetitive as they are presented in log values. In logscale, the control mean is equal to 0. We decided not to add the control bar to the graph.
The figure legends have been revised accordingly in line with the comments and the corrections made.
Additional corrections: in Tables 1 and 4 we have removed columns where data were unavailable due to undetermined gene expression.
Additional corrections
In the Abstract, one sentence has been removed because of repetition.
The proofreading corrections and the English text have been reconciled. You can see this by making changes with Track changes.
We have made corrections in accordance with the reviewers' comments. The comments were important, and we believe that the corrections really improved the publication. We are grateful to the reviewers.
Sincerely,
Donatas Stakišaitis
Round 3
Reviewer 1 Report
Comments and Suggestions for Authors
Dear authors, as you have already shown the tables, I don't think it's necessary to show the bar graphs. Besides, for the control group, the ΔΔCT =0, and 2-ΔΔCT =1. Please correct this content.
Author Response
Replies to the Reviewer's comment
Reviewer note
Dear authors, as you have already shown the tables, I don't think it's necessary to show the bar graphs. Besides, for the control group, the ΔΔCT = 0, and 2-ΔΔCT = 1. Please correct this content.
Answer
The bar graphs (Figures 1–3) were removed. According to the note, we added the ΔΔCT, 2-ΔΔCT mean values with SD for the control groups in Tables 2, 3, and 5.
The related sentences were removed from the text after the graphs were removed. Thank you for your comment.
Sincerely,
Donatas Stakišaitis